# Using Unmanned Aerial Vehicle Remote Sensing and a Monitoring Information System to Enhance the Management of Unauthorized Structures

**Yuanrong He [1,2], Weiwei Ma [1,*], Zelong Ma [1], Wenjie Fu [1], Chihcheng Chen [3], Cheng-Fu Yang [4,*] and Zhen Liu [3]**

[1] Big Data Institute of Digital Natural Disaster Monitoring in Fujian, Xiamen University of Technology, Xiamen 361024, China; 2012112001@xmut.edu.cn (Y.H.); zelongma123@gmail.com (Z.M.); fwj@ptu.edu.cn (W.F.)

[2] Key Laboratory of Ecological Environment and Information Atlas Fujian Provincial University, Putian University, Putian 351100, China

[3] School of Information Engineering, Jimei University, Xiamen 361021, China; 1761000018@jmu.edu.cn (C.C.); 201911810003@jmu.edu.cn (Z.L.)

[4] Department of Chemical and Materials Engineering, National University of Kaohsiung, Kaohsiung 081168, Taiwan

* Correspondence: maweiwei1019@gmail.com (W.M.); cfyang@nuk.edu.tw (C.-F.Y.)

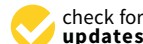

**Featured Application: Unauthorized construction is common in residential areas and can result in various types of harm, such as pieces of buildings falling off in bad weather and causing damage. These incidents may lead to injury or death, as well as structural damage, increasing financial loss to governments and negatively affecting the local population's livelihoods. In this study, we investigated a method that uses an unmanned aerial vehicle with remote sensing as well as a monitoring information system to enhance the detection and prevention of unauthorized construction.**

**Abstract:** In this research, we investigated using unmanned aerial vehicle (UAV) photographic technology to prevent the further expansion of unauthorized construction and thereby reduce postdisaster losses. First, UAV dynamic aerial photography was used to obtain dynamic digital surface model (DSM) data and elevation changes of 2–8 m as the initial sieve target. Then, two periods of dynamic orthophoto images were superimposed for human–computer interaction interpretation, so we could quickly distinguish buildings undergoing expansion, new construction, or demolition. At the same time, mobile geographic information system (GIS) software was used to survey the field, and the information gathered was developed to support unauthorized construction detection. Finally, aerial images, interpretation results, and ground survey information were integrated and released on WebGIS to build a regulatory platform that can achieve accurate management and effectively prevent violations.

**Keywords:** unmanned aerial vehicle (UAV); dynamic digital surface model (DSM); geographic information system (GIS)

## 1. Introduction

Over the past 40 years since China's reform and opening up, the pace of urban construction has dramatically increased [1]. Between 2000 and 2005 alone, the total area of China's provincial capitals increased by 90.15%. Between 2000 and 2010, China's urban area grew by 101.04% [2,3].

At the same time, the volume of unauthorized construction has also increased in direct proportion, which has seriously affected urban management and increased the workloads of urban supervision departments. Those engaging in unauthorized construction have not obtained the required permission for land use, structural planning, and construction. Since the promulgation of the urban planning law, unauthorized construction is regarded as illegal building [4]. Unauthorized structures such as corrugated iron houses in coastal cities are disastrous when a typhoon strikes, as the high winds cause the iron sheeting to fly through the air, destroying vehicles, shattering glass, and leading to other adverse consequences, including injuries and deaths. Quickly and effectively identifying unauthorized construction has become a key problem for urban planning departments. To identify illegal changes to buildings in urban areas, high-resolution aerial images can be useful for clearly showing the basic external characteristics of structures.

In the past, many methods had been investigated to detect changes in buildings. Matikainen and Hyyppa presented a method to automatically detect alterations to buildings using airborne laser scanner and digital aerial image data [5]. In a different approach, Huertas extracted data on buildings in old images and put the information into a database created using Site Model; new images were then compared with the models in the database to identify changes [6]. Agrostis applied an algorithm to a geographic information system (GIS) database to detect changes in buildings. The important core of the algorithm was measurement of the minimum square distance between original building boundaries and a vector polygon from the GIS in new images [7]. Watanabe used aerial images from different shooting angles and at different times. By eliminating the influence of parallax and building shadow caused by different shooting angles, the corresponding roof of a building in the two images was scanned to detect variation [8]. The buildings' change detection can also be used in different areas. For example, Vu used the specificity of LIDAR data to detect building changes in earthquake-prone areas [9].

Recently, Marin proposed a new method for employing multitemporal very high resolution (VHR) synthetic aperture radar (SAR) images to detect changes, using the regular back-scattering characteristics of buildings to detect new or completely demolished buildings [10]. Bouziani used existing geographic databases and prior knowledge to automatically detect building changes in urban environments from high-resolution images, thereby identifying illegal construction and monitoring urban growth [11]. Tian proposed a solution to monitor changes in buildings based on height variations in stereo imagery and digital surface models (DSMs). The change was detected via a stereo-matching methodology that used the joints of height changes and Kullback-Leibler divergence similarity measurements between original and later images [12]. Guo used a multilevel random forest (RF) classifier to classify VHR images and compared the changes in two datasets to detect building alterations, using GIS data [13]. Although the above-mentioned studies have achieved good results in the identification of unauthorized construction, they cannot precisely prevent unauthorized construction. Thus, it is necessary to find a method of effectively detecting and managing unauthorized construction. In this study, we took the streets in the Siming District of Xiamen City as a research target, and we used multiperiod data and images from an unmanned aerial vehicle (UAV) with remote sensing to generate a database. We then developed an initial screening tool that gathered DSM data from UAV remote sensing to automatically recognize changes in building elevation [14]. This was used to compare two orthogonal images in an interactive way to extract boundaries that indicated new construction, or the extension or demolition of buildings. When the tilt photographs of oblique buildings and 720° panoramic aerial photography were used simultaneously to manually verify the automatic extraction of results, we built a database of illegal buildings. In this study, the constructed supervisory systems included 3D visualization of important and key zones, the buildings' monitoring application software (APP) of named "dynamic inspect", and a WebGIS supervisory platform. We were thereby able to create a completely new supervision model that uses "seeing from the sky", "managing on the ground", and "searching on the Internet" to detect illegal building [15].

## 2. Identification and Supervision of Unauthorized Construction-

Ground object recognition and change detection based on remote sensing images constitute a complex process, especially for small targets such as buildings. Low-resolution remote sensing images are insufficient for extracting the features of ground objects and hence less useful for detecting changes in buildings to identify unauthorized construction. The high-resolution data sources used for change detection have some limitations; for example, it is difficult to obtain images from the QuickBird satellite, and high-resolution aerial images take large amounts of memory to store. At present, although some cities have carried out building change detection and research to identify illegal buildings, and have tried to establish a business application system for finding illegal buildings, there still is no complete theoretical and technological system for identifying illegal buildings, because these applications have not been widely promoted [7,16].

The basic idea of identifying illegal urban construction activity in this manner is to take the distribution of urban buildings as the monitoring object, then use UAV remote sensing technology to quickly obtain images and spatial distribution information for buildings through image processing, pattern recognition, and other methods. The results are then overlaid with planning map data that only experts can interpret, allowing managers to monitor current building information and identify unauthorized construction. This method can effectively deter unauthorized construction activities and aid in the supervision of overall urban planning. It also can provide high-quality, intuitive spatial information support for dealing with urban disasters and emergencies, and promote the sustainable development of urban planning and construction in China.

In this study, using UAV remote sensing for the dynamic monitoring of construction and the collection of monitoring information involved three main modules.

The first was the data processing and illegal building identification module. The aim of this module was to complete data acquisition and illegal building identification. High-resolution UAV remote sensing images were captured from the research objects. At the same time, automatic preliminary screening of changes in building height was carried out based on Model Builder in ArcGIS, and the changes were confirmed by manual visual interpretation. Then, unauthorized construction was verified using a 3D model and 720° panorama imaging to obtain accurate data on illegal buildings.

In the second module, we handled the data based on the WebGIS network information management system. This provided a dynamic supervision platform for a series of functions: target image identification, field verification, investigation and penal process monitoring, and historical tracing. We employed the architecture of WebGIS in the OpenGeo Suite and related technologies to analyze the design and integration process of PostGIS based on an object-oriented spatial database. By building a complete prototype system, it is possible to use WebGIS technology for digital information management related to unauthorized urban construction.

In the third module, we used a mobile grid management terminal system for data communication. This mobile intelligent terminal provided information support for daily operations so data could then be collected and analyzed in a timely manner for presentation to grid managers.

The built detection and management platform is shown in Figure 1.

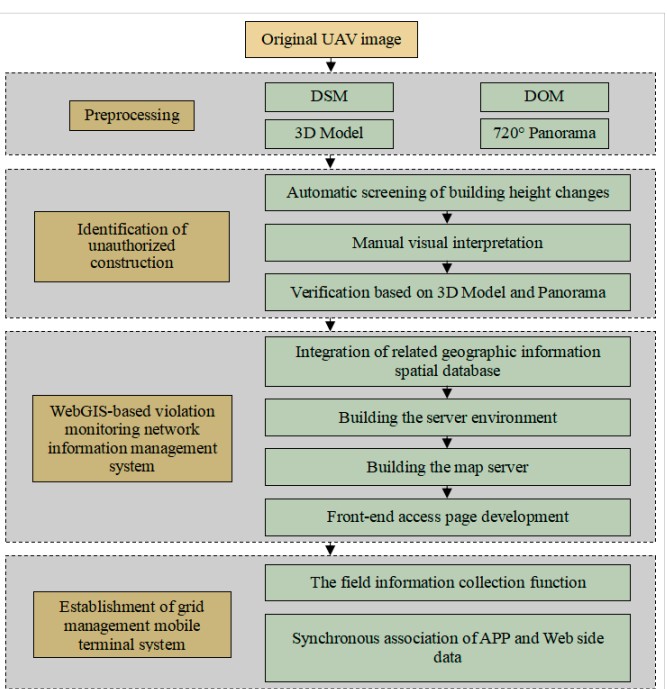

**Figure 1.** The built detection and management platform. DSM: digital surface model; DOM: digital orthophoto map

## 3. Aerial Images Capture and Data Processing

### 3.1. Preimage Data Acquisition

The first aerial images obtained from the research target were captured from 19 to 22 February 2017. The actual working time was 3 days, and the image spatial resolution was better than 10 cm. The acquired image data were processed into an orthophoto map by software 32 Gbit capacity. As ArcGIS software was used for image processing, we were able to add place names and administrative boundaries to the images, enabling us to make initial grid blocks according to streets and the jurisdictions of community organizations. After that, we divided the image data into 81 areas to make secondary grid blocks, then printed out 82 maps, one of which was of the total area. The map scale for each block was about 1:1000. These results could be used by the grid managers to "see from the sky".

### 3.2. Image Data Network Sharing

The volume of data had the potential to create bottlenecks that would slow down performance in the computer and at the interface with the GIS software, so we needed to publish on the web. Managers used a web browser or mobile phone to view aerial photos of big-data results. The web platform mainly enabled basic data input, grid data management, grid member management, daily operational supervision of grid personnel, and statistical analysis of events in the grid and of management functions in the background. Based on high-resolution aerial images of the GIS visual integrated development environment, we were able to effectively improve the ability to manage integrated grid monitoring data and provide an effective online technology for regulating unauthorized construction.

## 4. Identification of Unauthorized Construction

The second set of aerial images obtained from the research target was captured on 3 March 2017. Using aerial imagery postprocessing software, we generated a digital orthophoto map (DOM), DSM, 3D images, and other data. With this approach, as long as the DSM of the building is obtained and changes in the DSM are detected by comparison of images from the two dates, preliminary filtering can be done by computer. On the basis of preliminary computer selection, supplemented by manual

visual inspection, a 3D image and a 720° panoramic view can provide artificial verification, and then the unexpected stronghold can be obtained and the corresponding thematic maps can be produced.

*4.1. Preliminary Automatic Screening of Unauthorized Construction*

In this paper, the program was coded in several steps for preliminary identification of illegal buildings, and these main steps are shown in Figure 2. First, the projection transformation of preprocessed DSM model data was required in order to make DSM data with different phases have accurate superposition. Second, different DSM data with different phases were calculated by using ArcGIS built-in grid calculator. Third, if the difference value was smaller than the set threshold one, it was judged as a non-illegal building, otherwise it was an illegal building. And fourth, the result data of the building area was extracted by superimposing the masking data with the data of the building area.

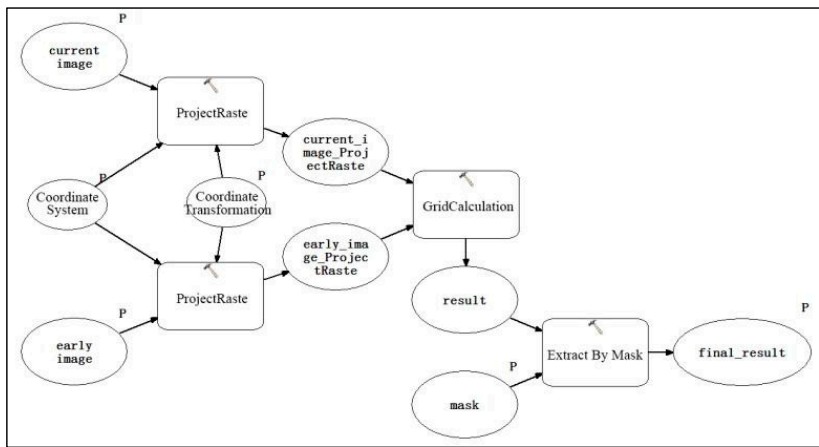

**Figure 2.** Automatic extraction flow chart of building elevation change detection.

The term "building changes" refers to new construction, capping, or demolition of buildings within a certain period of time in a specific area. These changes are mainly reflected in building height, as the results in Figure 3a,b show. By using the DSM model of a long time series in the observation area and combining this with a spatial analysis algorithm, we extracted a change distribution diagram of regional building height, as shown in Figure 4a. Further analysis of this diagram enabled us to set a reasonable number and interval of height difference classification values. Finally, we obtained a reclassification diagram of changes in buildings' heights. In combination with actual observation, as the buildings were constructed at different time differences and their heights were less than 1.5 m, they could be recognized as non-illegal areas, so such data were filtered out. The final result is shown in Figure 4b by selecting different symbol styles for the appropriate layers. The results in Figure 4 show that we can accurately and intuitively present dynamic changes in buildings' heights within a project area for a specific time period. By rapidly recognizing and filtering changes in buildings in a monitored region, including new buildings, illegal construction, and demolition, we can effectively reduce the effort required for early artificial visual discrimination of objects that can provide clues for further artificial visual discrimination.

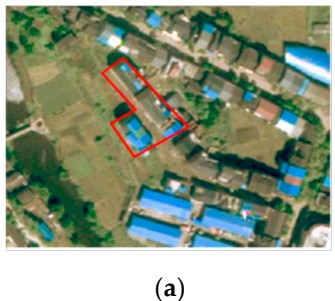

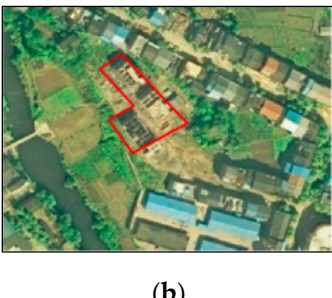

(**a**)　　　　　　　　　　　　　　　　　　　　　　(**b**)

**Figure 3.** Identification of (**a**) unauthorized construction and (**b**) undiminished unauthorized structures.

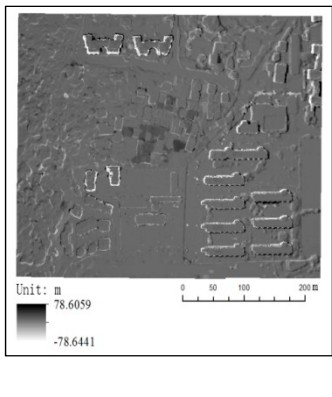

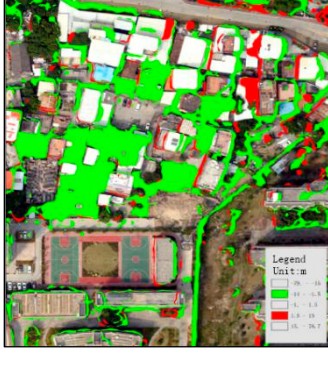

(**a**)　　　　　　　　　　　　　　　　　　　　　　(**b**)

**Figure 4.** Identification of building height: (**a**) change distribution map and (**b**) building height variation reclassification.

First, we pretreated the multiphase DSM model data for the same research area, including data exploration, projection transformation, and so forth, to ensure all the pretreated time-phase data was founded on the same accurate coordinate system for subsequent spatial analysis. Next, we used a grid computing tool to subtract the pre- and postprocessed phase data [17]. The formula is as follows:

$$\text{Result} = \text{current-image ProjectRaste} - \text{early-image ProjectRaste} \tag{1}$$

where current-image ProjectRaste is the post-temporal DSM model (representing the current state), and early-image ProjectRaste is the antecedent phase DSM model (representing the past state). The difference between the early-and current-phase DSM model can be displayed for the same region. A positive result indicates an increase in height, which may be due to new construction or illegal buildings. A negative result indicates a lower elevation, which may be the result of demolition. We can thereby obtain a building height distribution map.

Through the above process, we achieved automatic extraction of all elevation changes in a certain region during a certain time period. As the main objective of this project was to examine built areas, potential disturbance factors such as woodland, shrubs, and mountainous areas were eliminated, and we applied masking tools to the built area. A difference in phase height less than ±1.5 m was considered a nonbuilding change interval and was filtered out. The test data were then reclassified at intervals of 1 and 1.5 m. A distribution map of building height change was thus obtained and a data processing model was created to generate the appropriate data processing tools. The tools can be shared for further development and improvement. It also has a good interface to facilitate bulk data processing.

After computer-automated processing, new DSM data were obtained: elevation differences in two-stage images. For convenient application, the data's attributes can be classified using a system in which a threshold value is assigned a color for the range area. With that, managers can see a lot of

extracted patches on the preprocessing images. Visual comparison of base maps allows managers to see elevation changes in the houses of a given area. The ArcGIS recognition function is then used to check the buildings' elevation and guide the follow-up visual interpretation work. A thematic map of preliminary screening results can also be made, as Figure 5 shows.

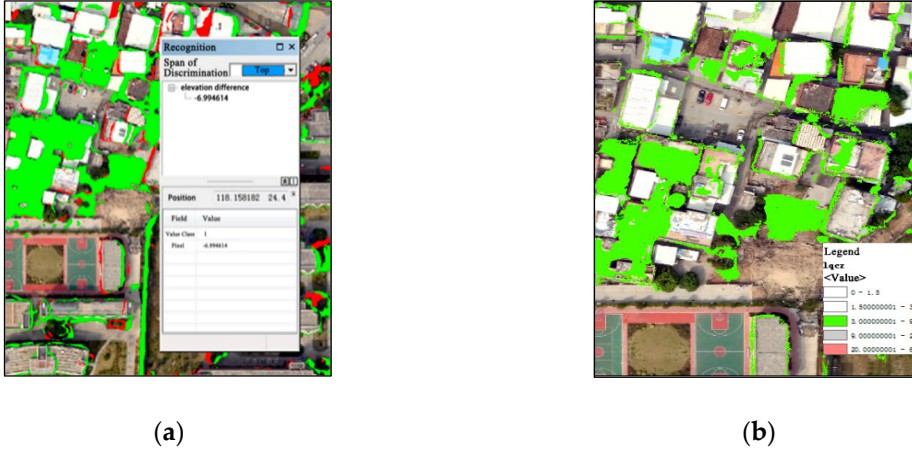

(**a**)                                                                  (**b**)

**Figure 5.** (**a**) Patches elevation query and (**b**) patches elevation drawing.

### 4.2. Artificial Visual Interpretations

Next, we established the database and the new corresponding elements, including the illegal and demolition buildings. Computer processing enables images to be extracted when obvious addition or removal of buildings has occurred, but the variations of floors in buildings (the blocks shown in Figure 6) are not easily visible in the image. The elevation of the DSM chart can be checked using ArcGIS recognition to confirm architectural changes have occurred. Figure 6 shows some results of artificial visual interpretation.

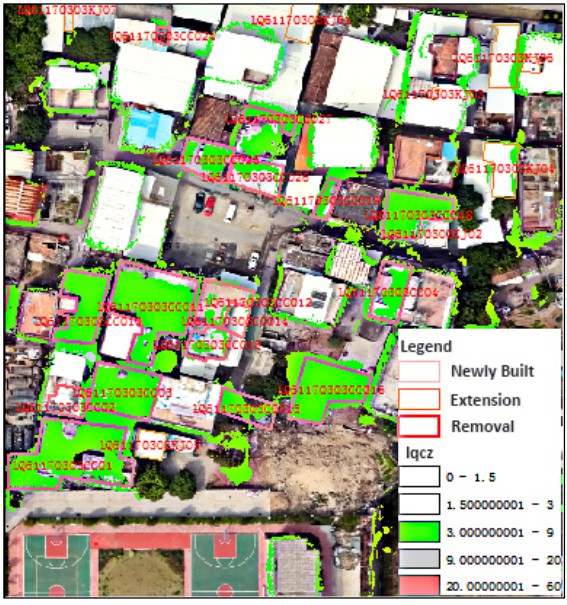

**Figure 6.** Example of artificial visual interpretation results.

*4.3. Two Kinds of 3D Verification*

4.3.1. Three-Dimensional Image Verification

Oblique photographs of the aerial images were obtained using postprocessing software, enabling us to create 3D images. With a 3D image model, we could take measurements of buildings, such as height, area, and so on, as Figure 7 shows. The oblique photographic data were then output into open scene graph binary (OSGB, one of the formats for oblique photographic data) format for 3D data; this is an open-source format that can support the secondary application of a variety of 3D software. Managers could then release the exported OSGB format data to the online map browser named "Local space". Publishing this information online would enable inhabitants to identify unauthorized construction. This method could deter illegal activity and at the same time facilitate the handling of lawful judgments, which could be used to educate people about legal procedures and consequences related to illegal buildings.

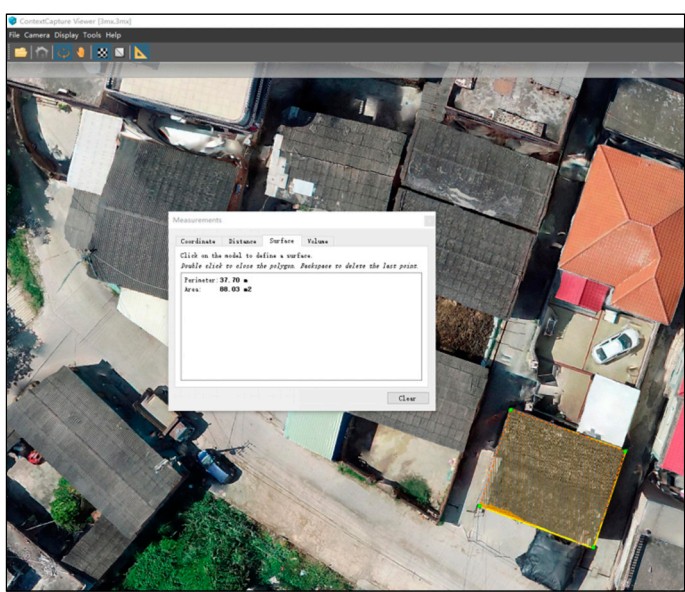

**Figure 7.** Building measurement based on 3D model.

4.3.2. 720° Panoramic Verification

A 720° panorama is achieved by combining horizontal 360° and vertical 360° images. The shooting and production speeds are very fast. Images are then collected as data from a website, and a web browser can clearly and intuitively display the results from these images. In general, the results are viewable quickly and easily on a mobile phone or a PC [18,19].

*4.4. Verification of Precision Extraction*

In this paper, the automatic extraction program of "Building Elevation Change" was used to extract the suspected illegal building spots on the map with height changes larger than 1.5m. Based on the extraction data, we could use 3D images and panoramic images to verify the suspected illegal construction spots, and then we could obtain the illegal construction spots for assignment. This is a human–computer interaction verification process. For that, we took Siming District of Xiamen City as the experimental object to verify the accuracy. According to the field verification, 27 illegal buildings could be found in this block. A total of 26 suspected illegal buildings spots on the map were extracted in this experiment, 21 of which were actual illegal buildings, 5 of which were misjudged, and 6 of which were lost. The accuracy of human–computer interaction extraction of illegal buildings was 78%, which can meet the managers' requirements.

## 5. WebGIS-Based Violation Monitoring Network Information Management System

The main method for managing illegal building is to use a regulatory platform to supervise a series of functions, such as target image recognition, field verification, investigation and process monitoring, and historical tracking of illegal buildings. Therefore, the adopted digital information management system needs accurate analyses of positioning and spatial information about illegal entities, which are the main functions of the GIS. Compared with a traditional GIS software platform, the system platform built with WebGIS can access data from any platform and any place. The user-friendly interface can be customized with simple, clear, and intuitive images. It can effectively balance a high volume of spatial data for processing calculation and maximize the utilization of computer resources. It can also use a spatial database for data management and effectively improve efficiency and accuracy.

### 5.1. System Functions and Design

According to the actual demands of illegal construction management, the system is divided into the following four functional modules, as shown in Figure 8.

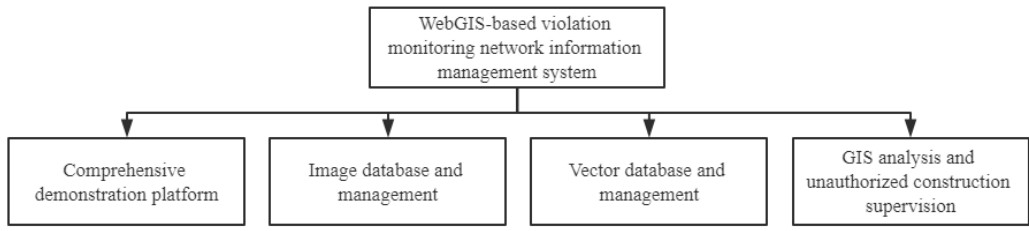

**Figure 8.** Functional modules of designed system.

(1) Comprehensive demonstration module

Based on remote sensing data of UAV, every illegal construction project can be processed into the products of orthophoto images, digital surface models, 3D models, and panoramic images. These products can display the inspection area in a more comprehensive display mode and show the development status of illegal buildings in a more vivid and flexible way. It is one way of mixing management and supervision for the phenomenon of illegal buildings.

(2) Image database and management module

The main task of this module is to manage and schedule the preprocessed UAV remote sensing images and it can provide basic services for the entire management system. It mainly includes the establishment of the image database, the import of multiformat image data, the browsing and displaying of image data, and the location and scope of image data, and other information management.

(3) Vector database and management module

This module is mainly used for the management and scheduling of the relevant vector data generating from the processing of original data and vector data of buildings' planning provided by relevant departments. By superimposing the vector data on the images and comparing them with the planning approval maps, they can accurately identify the illegal buildings.

(4) GIS analysis and monitoring module of illegal buildings

By using the accurate positioning and analysis capability of spatial information of the GIS system, the information such as the location, area, and center point of the illegal buildings can be calculated. Then, we can exclude and screen out the initially extracted illegal buildings' information, and we can determine the geographical locations of the illegal buildings, the types of the illegal buildings, and other information to prepare for the next step of monitoring. The monitoring work of illegal buildings is mainly divided into six steps: segmentation grid, polygon establishment, polygon confirmation,

task distribution, field verification, and archive confirmation. The monitoring area is gridded and divided, and the management polygon of illegal buildings is created (a polygon represents one household of illegal building). After confirming that there is no error in the polygon information, the task is dispatched to the mobile station's grid members, and the grid members can verify the illegal construction in the field, confirm the relevant information, and then return the data through the mobile terminal to archive.

### *5.2. System Construction*

This section uses the architecture of WebGIS present in OpenGeo Suite and related technologies to analyze the design and integration process of PostGIS based on an object-oriented spatial database. In other words, we elaborate on how to comprehensively use a WebGIS-based prototype for analyzing a digital information management system to detect unauthorized construction [20,21].

### 5.2.1. System Architecture and Development Technology

At present, the WebGIS system mainly uses the browser/server (B/S) mode. The B/S mode simplifies client software, develops the system's functions, and maintains and updates recognizing methods on the application server. The database server is composed of a three-layer architecture, and the constructed data is placed in the server's database to form a client layer and the middle application layer. Compared to the B/S mode and the traditional standalone client mode, it is easier to manage and maintain, makes the least demands on the client, and is convenient to use and popularize. WebGIS implementation in this study was based on the development of the open-source software OpenGeo Suite, which provides a comprehensive web-based mapping and data-sharing solution. Software development was done mainly with GeoServer, PostGIS, OpenLayers, and other components. In addition, OpenGeo Suite provided other components, such as GeoExplore and GeoWebCache. When OpenGeo Suite was being used, the "Dashboard" was the platform for logging in and editing. Figure 9 shows the software's architecture.

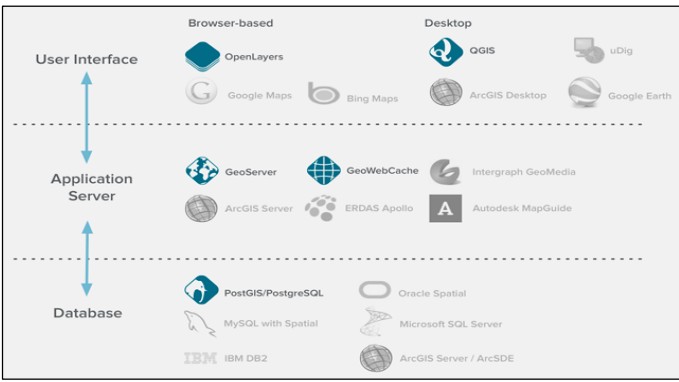

**Figure 9.** Hierarchical relationship of the OpenGeo Suite series software.

### 5.2.2. Spatial Database Design and Integration

In the process of designing a spatial database, two aspects require attention: the building-related geographic information data, such as the UAV aerial image data, and the preliminary screening vector data and related basic data such as point of interest (POI). At the same time, the landscape architecture data needs to use the management of spatial database for conducting the analyses of detailed needs. Relevant departments will construct a standard mode that suits their processing system to achieve the dynamic monitoring of unauthorized construction. To achieve these goals, these steps should be followed in designing the database:

(1) Conceptual model of spatial data: Spatial data classification includes vectors and grids. A three-level approach to the data model is required, namely, conceptual model, logical model, and

physical model. Traditional relational databases cannot be used directly to manage spatial data; instead, data models and methods must be extended. The conceptual model is the first level of abstraction from the real world to the machine world.

(2) Logic model of spatial data: A logical model can be understood by mapping the real world to a computer or database system. In such a model, the database areas are: emerged hierarchical model, mesh model, relational model, and object-relational model. This article mainly relates to the object-relational model, which results from combining relational database technology and object-oriented programming. An object-relational database generally has the following functions: ① extend data types, ② support complex objects, ③ support the concept of inheritance, and ④ provide common rules and systems. PostGIS is based on predefined data types and extended geometry types. Predefined data types are stored and managed using existing data types in relational databases, whereas extended data types are stored and managed using extended geometry data types.

### 5.2.3. System Development and Implementation

(1) Building the server environment: Establishing a server environment mainly involves the installation of OpenGeo Suite, a PostGIS database, and QGIS. All the software needs to be installed in the same directory. After the database is constructed, users need to load the functions of the extension module provided by PostGIS, which are mainly a series of processing functions based on the OpenGIS standard. After connection to the database, the server environment is realized by opening the "console" box in the PgAdmin menu and then executing the following SQL statement:

- Enable PostGIS (includes raster) CREATE EXTENSION postures;
- Enable Topology CREATE EXTENSION postgis_topology;
- Fuzzy matching needed for Tiger CREATE EXTENSION fuzzystrmatch;
- Enable US Tiger Geocoder CREATE EXTENSION postgis_tiger_geocoder;

(2) Building the map server: GeoServer structures mainly use the default Jetty application server of OpenGeo. Because GeoServer is built on Java 2 Platform Enterprise Edition (J2EE), users need to install and configure the Java environment before installing OpenGeo. They also need to start Geoserver-related services after installation, then open OpenGeo's Dashboard console through a browser.

At the same time the needed vector data are imported and connected with GeoServer and PostGIS database, the user selects Import Data-> postGIS and enters the correct parameters, including the database name, port number, user name, password, and so on. After the data import is completed, the user can employ GeoExplorer to modify the layer style, including color, symbol, and transparency. Multiple layers can further build the layer group to improve computational efficiency, as shown in Figure 10.

For importing image data, GeoServer provides an extended model that can directly import the data into the remote sensing images and automatically generate a mosaic. So, to acquire UAV aerial data, the user can go directly along the path (Import Data-> Mosaic) to select the appropriate storage path for completing the data import. After the import is completed, the user can employ the GeoWebCache tool to set the appropriate parameters, including hierarchy levels and image formats. After an image pyramid is built, the user can preview it, as shown in Figure 11.

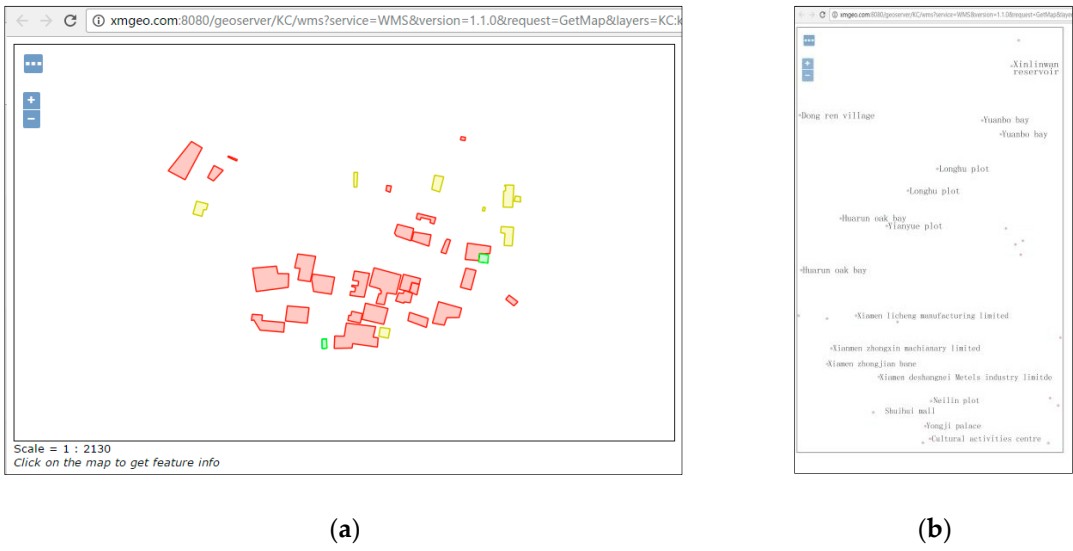

(**a**)                                                                    (**b**)

**Figure 10.** Layer modification of (**a**) the vector layer group and (**b**) the label layer.

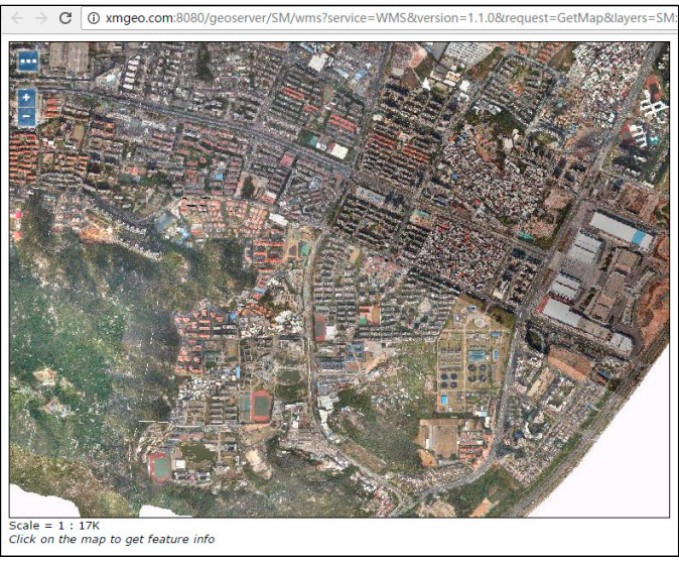

**Figure 11.** Image release.

(3) Front-end access page development: In this study, we used the OpenLayers 3 and LeafLet frameworks to develop the front page. When the remote sensing data for the demonstration of research target cases is introduced, the page results show that the core part of the Map display is the definition of the Map element. Then, the remote sensing data can define the current map needed to load the layer. In this method, requisite points of interest, roads, and other layers can be added to order. The display of different layers can be controlled through the layer's panel, and the final effect is shown in Figure 12.

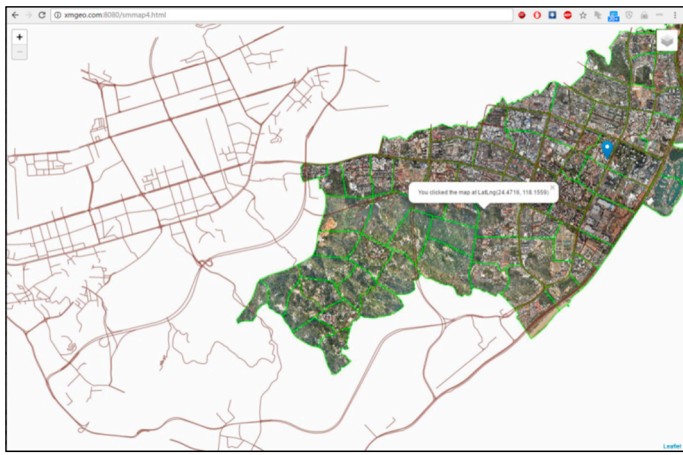

**Figure 12.** Vector layer release.

## 6. Grid Management Mobile System Construction

### 6.1. Function Design of Grid Management on Mobile Terminal System

As grid management has become routine, large-scale, and unevenly distributed, it is necessary to introduce mobile terminals as information support tools for daily operations, to prepare grid managers for timely on-site data collection and analysis. The system framework of the mobile law enforcement platform for illegal constructions mainly consists of three parts, namely, mobile terminal, PC terminal, and server. The functions of the mobile terminal include main functions such as home page, grid task, illegal construction record, location navigation, and opinion feedback. The PC terminal accesses the monitoring system through the browser, mainly for data management and display and task distribution and management of the mobile terminal, so that law enforcement personnel can carry out their work more easily. The server mainly serves mobile terminals, such as uploading evidence to the server, obtaining personal tasks, historical remote sensing images of illegal construction areas, and so forth. The physical structure of the system is shown in Figure 13.

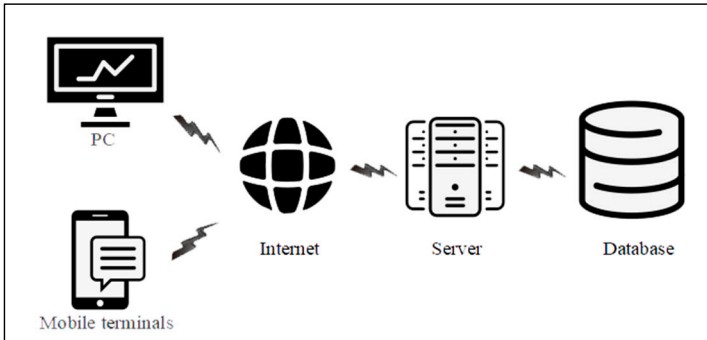

**Figure 13.** The physical structure of the system.

The main functions of the mobile terminal module include home page, grid task, illegal construction record, location navigation, and opinion feedback.

(1) Home page

The home page is a map that mainly shows the current location of the grid members and the surrounding illegal buildings.

(2) Grid task

The main function of the grid task module is to perform task notification. Through the network information management system of violation monitoring based on WebGIS, each grid member can receive the currently assigned monitoring tasks in the grid task module. As long as the grid member selects a polygon, he can use the positioning module, navigating himself to the task location.

(3) Illegal construction record

The illegal construction record module is investigated for on-site forensics. The module displays information such as the districts, geographic coordinates, areas, and types of the illegal buildings. At the same time, it will retrieve the historical images and current images of the illegal buildings in the server. The grid members can compare the images' information with the actual situation to determine whether it is illegal. Also, the grid members can use the module for photo samplings, data modifications, and other works.

(4) Location navigation

This module displays all illegal buildings' positions within the grid members' management area. By selecting a location, the navigation function can be provided, so that the grid members can find the illegal buildings more quickly and improve work efficiency. At the same time, the positioning navigation module will transmit the current positions of the grid members to the PC terminal in real time, and the management personnel can manage the current work progress of the grid members.

(5) Opinion feedback

This module is for users who have experience of the mobile law enforcement platform to put forward suggestions for improvement to the developers.

*6.2. Function Realization of Mobile Terminal*

In this paper, the HTML5 + PhoneGap framework was adopted to implement the development of an APP on the user's side, and at the same time, back-end logic computing service was provided through J2EE software, which not only improves the stability but also takes into account the user's habits and updates the service accordingly.

6.2.1. Illegal Building Information Verification Preparation

Before government staff enter community buildings to collect information, they need a general understanding of the architectural regulation area and familiarity with the terrain and the path of the query location. After confirming the location, they go there carrying the APP-related equipment (for example, cell phones) for grids' search. Using APP-based positioning to navigate, they can reach the specified location. When entering the work area, they then call out the correct projective images of the marked buildings and inspect the spots marked as having changed.

6.2.2. Mobile APP Site Information Collection

As the information gathered by the Grid Inspector APP includes the precise location of a household (including the specific house number) and the architectural changes observed (including photos and other evidence), sensitive personal information related to the vital interests of many people is collected. The system therefore requires that grid members use their own accounts' passwords to log in and start building information collection maps.

After each grid's operator logs in, they can view their inspection records to check their own inspection places and quickly find the nondetection places. This means that grid operators can have an accurate grasp of the historical records for their regions of responsibility, giving them a clear understanding of the changes in their regulatory regions, and then can effectively monitor to detect

unauthorized and illegal construction. These results prove that the mobile APP can help grid operators monitor newly built or illegal buildings and prevent dangerous demolitions, so operators can smoothly apply regulations and enforce laws.

Grid operators' daily tasks will be to reach the grid site and use the APP to select grids, times, and statuses, then enter selective auxiliary information. Via photo records, they can also add corresponding notes. Because the system provides geographical locations and positional navigation, grid operators can arrive at the correct positions at the right time to collect information. Using the "inquiry on the ground" approach can greatly increase the operators' efficiency and the reliability and timeliness of data management. In addition, the system keeps track of grid staff's daily work, so histories and inspection results can be tracked through the web-based system and improve daily grid management.

### 6.2.3. Building Monitoring APP and Web-Side Data Synchronization Association

For information dissemination, monitoring and management terminals should provide mobile data interfaces so that grid operators can synchronize handling the processing steps and the data management system. This system can provide decision makers with real-time information from the web-based platform, helping to achieve the regular output of work reports. After grid members have collected site information and associated data, the managers can synchronize to discern relevant information and data, including the names and times of events, specific house numbers, photos, and events that have occurred on the site. Management staff can use the associated building information to determine whether action should be taken.

### 6.2.4. Illegal Building Query Management Systems

In this study, the management system summarized the DOM, DSM, and 720° panoramas, the measurement model using tilt photography, law enforcement records, and photo data to develop an office's network platform. This could then be used to check, in real time, the law enforcement records for a particular address. These results show that the investigated system can achieve the goals of "seeing from the sky", "managing on the ground", and "searching on the Internet", essential tools for stopping and preventing illegal construction.

## 7. Discussion

In this paper, the DSM data of multitemporal images were used to identify the elevation changes of illegal buildings with remote sensing data of UAV in different periods. With the excellent spatial information analysis ability and data management ability of WebGIS, this research constructed the network information management system for monitoring illegal buildings. By utilizing the timeliness and convenience of mobile terminal, the system was constructed and the grid management operation level of the system was improved. Table 1 compares the proposed method with other studies; when remote sensing data of UAV is used, the images with a data accuracy of 0.06 m/pix can be obtained, which is much better than that of satellite remote sensing images of 2.5 m/pix. The improvements of images' resolution are beneficial to the extraction information of illegal buildings and higher resolutions can lead to having higher extraction accuracy. In terms of the extraction methods of illegal buildings, this research started from the height determinations of the buildings, screened out the changes of the buildings, and then extracted the illegal buildings by human–computer interaction to ensure the accuracy of the illegal building extraction, but the level of automation was low. Through the acquisition and processing of dynamic aerial images, relative information releasing by network, grid management, and collection of relevant building information by mobile APP, the effective supervision of urban illegal buildings can be reached. A three-dimensional model monitoring system based on sky, ground, and Internet can provide strong technical supports for urban constructions.

**Table 1.** Comparison table of illegal buildings' identification and monitoring research. UAV: unmanned aerial vehicle

| Project | In This Paper | Others [22,23] |
|---|---|---|
| Basic data | UAV remote sensing data (0.06 m/pix) | Satellite remote sensing data (2.5 m/pix) |
| Convenience for remote sensing data | Easy | Difficult |
| Illegal building extraction method | Multitemporal elevation difference calculation, man–machine interaction | Post comparison of multitemporal image classification, change detection combining strategies at the pixel and the feature levels, building variation pixels were detected by k-means clustering technique |
| Accuracy of illegal buildings' extraction | 78% | 75% [23] |
| Dimension of illegal building supervision | Three-dimensional | Two-dimensional |

The identification and monitoring of illegal buildings is a complex task involving multiple disciplines and the resolution of various key issues, such as the corrections of images' geometry and radiation, feature extraction and description, knowledge expression and integration analysis of multisource data, and knowledge. Along with the support of basic theories such as computer technology, remote sensing technology, image processing, and pattern recognition, the geographic information system for the unified management of geographic information and other attributes' information has also developed rapidly. Based on these technologies, carrying on spatial information mining and knowledge discovery are the problems that need to be solved at present, and they are some of the hotspots and difficulties of GIS research. How to achieve high-level integration of existing technologies and methods to achieve more efficient work is also an important issue for further research.

## 8. Conclusions

In this study, we successfully used UAV remote sensing for the dynamic monitoring of construction and the collection of monitoring information. This involved three main modules: processing data and identifying illegal buildings; handling the data based on the WebGIS network information management system; and using a mobile grid management terminal system for data communication. The main method for managing illegal buildings was to use a regulatory platform to supervise a series of functions: target image recognition, field verification, investigation and process monitoring, and historical tracking of illegal buildings. When aerial imagery postprocessing software was used, we were able to generate a digital orthophoto map, a digital surface model (DSM), 3D images, and other data. On the basis of preliminary computer selection, supplemented by manual visual inspection, a 3D image and a 720° panoramic view provided artificial verification, then the unexpected stronghold was obtained and the corresponding thematic maps were produced. After computer-automated processing, new DSM data were obtained from elevation differences in two-stage images, and illegal buildings could be identified. The investigated algorithm achieved the goals of "seeing from the sky", "managing on the ground", and "searching on the Internet", essential tools for stopping and preventing illegal construction.

**Author Contributions:** Investigation, Y.H., W.M., Z.M., and C.C.; Methodology, Y.H., Z.L., C.-F.Y., and W.F.; Formal analysis, Y.H., Z.L., W.F., and C.C.; Writing—original draft preparation, Y.H., W.M., C.-F.Y., and Z.M.; Writing—review and editing, C.-F.Y., Y.H., W.M., and C.C.

**Funding:** This study was supported by Longyan City Science and Technology Plan Foreign Cooperation Project of China (Grant No.2018LYF7006) and Open Fund Programs of Key Laboratory of Ecological Environment and Information Atlas (Putian University) Fujian Provincial University of China (Grant No.ST17002). This work was also supported by projects under the Nos. MOST 108-2221-E- 390-005 and MOST 108-2622-E-390-002-CC3.

**Conflicts of Interest:** The authors declare no conflict of interest.

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
