# Peer review of "Using Unmanned Aerial Vehicle Remote Sensing and a Monitoring Information System to Enhance the Management of Unauthorized Structures"

_applsci, doi:10.3390/app9224954_

Round 1

Reviewer 1 Report

Please include details of the algorithms used to detect unauthorized construction of buildings. If would be valid to use canned methods/programs but at least details on how the programs work should be included in the discussion.  Authors need to compare their results to results obtained with other methods to see how their approach compares to other ones.

Author Response

Please include details of the algorithms used to detect unauthorized construction of buildings. If would be valid to use canned methods/programs but at least details on how the programs work should be included in the discussion. Authors need to compare their results to results obtained with other methods to see how their approach compares to other ones.

Answer: Thanks for the comment, we have added specific steps of the algorithm in the section of extraction of suspected illegal buildings and compared the research results with those of others in the discussion. Please see lines 164~171 and lines 480~507.

Reviewer 2 Report

The research article is focused on the Using Unmanned Aerial Vehicle Remote Sensing and a Monitoring Information System to Enhance the Management of Unauthorized Structures.

As the author describes in the introduction: "Between 2000 and 2005 alone, the total area of ​​China's provincial capitals 41 increased by 90.15%". This is a major global problem, but I alalyzed this problem is bigest of the concerns particular of the China’s provincial. That is why the research issue is in my opinion highly topical.

In the abstract section I miss the references to the similar research area. I also recommend adding a few references in this section.

I would like to ask the authors of the research article for a better explanation of the comparison of the three main modules involved in processing and identifying illegal buildings: WebGIS network management information management system; and using a grid management system for data communication.

For my opinion is necessary to better explain the accuracy of these methods of comparing the problems of Unauthorized Structures.

I recommend adding the part of the future research.

I wish the authors a lot of strength in further research on this issue.

Author Response

As the author describes in the introduction: "Between 2000 and 2005 alone, the total area of China's provincial capitals 41 increased by 90.15%". This is a major global problem, but I analyzed this problem is biggest of the concerns particular of the China’s provincial. That is why the research issue is in my opinion highly topical.

Answer: Thanks for reviewer’s agreement.

In the abstract section I miss the references to the similar research area. I also recommend adding a few references in this section.

Answer: Thanks for the comment, we have done more researches on the urban sprawl of China and added some valuable content and references in the introduction section, please see line 42 and references 2-3; lines 480-498 and references 23-24.

I would like to ask the authors of the research article for a better explanation of the comparison of the three main modules involved in processing and identifying illegal buildings: WebGIS network management information management system; and using a grid management system for data communication.

Answer: Thanks for the comment, the detailed design ideas and functions of these three modules have been supplemented. Please see lines 272~305 and lines 390~429.

For my opinion is necessary to better explain the accuracy of these methods of comparing the problems of Unauthorized Structures.

Answer: Thanks for the comment, we added the accuracy verification of extraction of illegal buildings. Please see lines 251~260.

I recommend adding the part of the future research. I wish the authors a lot of strength in further research on this issue.

Answer: Thanks for the comment. In the discussion section, we added the future research direction of illegal building identification and monitoring. Please see lines 498~507. We will do more in-depth research in the field of illegal building identification and monitoring.